# CircRNAs and RNA-Binding Proteins Involved in the Pathogenesis of Cancers or Central Nervous System Disorders

**DOI:** 10.3390/ncrna9020023

**Published:** 2023-03-31

**Authors:** Yuka Ikeda, Sae Morikawa, Moeka Nakashima, Sayuri Yoshikawa, Kurumi Taniguchi, Haruka Sawamura, Naoko Suga, Ai Tsuji, Satoru Matsuda

**Affiliations:** Department of Food Science and Nutrition, Nara Women’s University, Kita-Uoya Nishimachi, Nara 630-8506, Japan

**Keywords:** circular RNA, APRO family protein, RNA binding protein, translational regulator, poly(A)-binding protein

## Abstract

Circular RNAs (circRNAs), a newly recognized group of noncoding RNA transcripts, have established widespread attention due to their regulatory role in cell signaling. They are covalently closed noncoding RNAs that form a loop, and are typically generated during the splicing of precursor RNAs. CircRNAs are key post-transcriptional and post-translational regulators of gene expression programs that might influence cellular response and/or function. In particular, circRNAs have been considered to function as sponges of specific miRNA, regulating cellular processes at the post-transcription stage. Accumulating evidence has shown that the aberrant expression of circRNAs could play a key role in the pathogenesis of several diseases. Notably, circRNAs, microRNAs, and several RNA-binding proteins, including the antiproliferative (APRO) family proteins, could be indispensable gene modulators, which might be strongly linked to the occurrence of diseases. In addition, circRNAs have attracted general interest for their stability, abundance in the brain, and their capability to cross the blood–brain barrier. Here, we present the current findings and theragnostic potentials of circRNAs in several diseases. With this, we aim to provide new insights to support the development of novel diagnostic and/or therapeutic strategies for these diseases.

## 1. Introduction

Many studies have identified some non-coding RNAs (ncRNAs) with abnormal expressions in several diseases and/or disorders [1]. Among them, circular RNAs (circRNAs) are a special type of endogenous ncRNAs, probably formed by back-splicing events, which have attracted more and more interest nowadays. CircRNAs are closed loop structures [2], in which the 3′ and 5′ ends are covalently joined [3]. The altered expression of specific circRNAs might play an important role in human diseases and/or disorders [4]. CircRNAs were first thought to be something similar to viroids in plants [5]. Although the abundance of a number of circRNAs is lower than their counterpart linear RNAs, circRNAs are commonly definitely expressed [6]. In addition, circRNAs are much more stable than linear mRNAs or non-coding microRNAs (miRNAs) [7]. The lack of a linear terminal at both the 3′ and 5′ ends might hamper their degradation by RNases, which could increase their stability in the extracellular environment, and support their helpfulness as biomarkers of diseases [8]. In fact, it has been shown that circRNAs are more resistant than linear mRNA to the degradation of RNase R due to the lack of 5′ and 3′ ends, including the terminal 5′ caps and/or 3′ poly (A) tails [9]. CircRNAs could play important roles in cellular processes. The first mechanism, with regards to the role of circRNAs, might be as a sponge of miRNAs [10]. Some abundant circRNAs could associate with miRNAs through regions of complementarity, which may enable the translation of the mRNAs by taking the miRNAs away from the target mRNAs. In addition, circRNA could also bind to proteins in the related signaling pathways [11]. Furthermore, circRNAs could be translated to protein peptides [12]. Consequently, circRNAs might promote RNA translation and/or the stabilization of the miRNA assembly. Therefore, the specific expression of circRNAs may be different from the other classes of RNAs.

According to their splicing sequence, circRNAs can be categorized into several groups. Exon–intron circRNAs and intronic circRNAs are mainly located in the nucleus, suggesting that they may be involved in gene expression [13]. Exonic circRNAs are the most abundant circRNAs found in the cytoplasm [14]. A lariat structure may be formed by the covalent binding of the splice donor of the downstream exon of the precursor mRNA and the splice acceptor of the upstream exon, leading to the formation of exon–intron circRNAs and exonic circRNAs. In this model, further back-splicing by the covalent joining of the 3′ and 5′ ends might result in an intronic circRNA lariat. In some cases, exon skipping might also result in mixed circRNA lariats [15]. Circularization might be created from the introns flanking the exons of the pre-mRNA sequence. In addition, it has been reported that the biogenesis of circRNAs could be regulated by RNA-binding proteins (RBPs) [16]. A circular structure could be created by the connection of RBPs to introns on both flanking exons of the pre-mRNA sequence. RBPs could recognize the specific motifs of the back-splicing to form circRNAs. CircRNA may be localized in the nucleus, where it can recruit several proteins to modify the chromatin structure and/or to bind to DNA forming an RNA–DNA hybrid for the transcriptional alterations [17]. In the cytoplasm, circRNAs have also been detected; regulating target mRNA expression by acting as miRNA sponges [18]. CircRNAs are broadly present in tissues, blood, and urine with structural stability [19]. Similar to the ncRNAs, circRNAs could be encumbered into exosome vesicles to facilitate cell–cell communication [20].

Again, circRNAs have emerged as novel regulators of gene expression by sequestering miRNAs and RBPs [21]. In addition, it has been suggested that circRNAs might play a critical role in regulating cellular events by interacting with RBPs [22]. In cases of miRNAs, miRNAs could regulate translation and mRNA stability by binding target mRNAs in a complex with Argonaute (AGO) proteins [23]. AGO proteins might interact with a member of the trinucleotide repeat containing six (TNRC6) family proteins to form a microRNP complex, which recruits the carbon catabolite repression 4 (CCR4)-negative on the TATA-less (NOT) complex to accelerate deadenylation with the inhibition of translation [24]. Deletion of the poly (A)-binding protein (PABP) interacting motif (PAM2) from the TNRC6 could abolish the translational activation, suggesting the involvement of PABP in the functional process of circRNAs and/or miRNAs [24]. Interestingly, the transducer of erbB2 1 (Tob1), a member of the antiproliferative (APRO) protein family, could simultaneously interact with the poly (A) nuclease complex CCR4-chromatin assembly factor-1 (CAF1) and the cytoplasmic PABP [25]. In addition, the transducer of erbB2 2 (Tob2), another member of the APRO protein family, could promote deadenylation by recruiting Caf1 deadenylase onto the mRNA poly (A) tail by also interacting with PABP [26]. The APRO family genes have been categorized in the group of immediate early growth responsive genes [27]. The gene products might include similar molecules including pheochromocytoma cell-3(PC3)/tetradecanoyl phorbol acetate-inducible sequences 21 (TIS21)/B-cell translocation gene 2 (BTG2), B-cell translocation gene 1 (BTG1), Tob1, Tob2, abundant in neuroepithelium area (ANA)/B-cell translocation gene 3 (BTG3), PC3B and others [27]. These APRO family proteins have been described as being involved in diverse human diseases including cancer [28]. CircRNAs may also conceivably provide a layer of regulation in protein synthesis and/or in diverse human diseases [29]. In addition, circRNAs may bind to mRNA directly to drive translation, or may play a key role in the regulation of alternative splicing [30]. Translational control may be a crucial component of tumor cell survival, cancer development, and/or cancer cell progression [31]. In general, the translation might be initiated from the circularization of mRNA and the binding of PABP onto the poly (A) tail and eukaryotic translation initiation factor 4 G (eIF4G) on the 5′ cap of the mRNA translation initiation complex in the cytoplasm. Therefore, blocking the interaction of PABP and eIF4G could prevent the start of translation and the following important protein synthesis [32].

## 2. Roles of CircRNAs in Pathophysiology

In the nucleus, circRNAs can be synthesized by the reverse splicing of coding exons and/or mRNA splicing without degradation of the intron [33]. CircRNAs can also regulate gene expression in the nucleus by binding to miRNA or proteins [34]. Interestingly, many studies have confirmed the role of circRNA in various diseases. For example, circRNA homeodomain-interacting protein kinase 3 (HIPK3) may induce endothelial proliferation and/or vascular dysfunction in diabetic retinopathy [35]. Cancer-specific circRNAs could also promote transformation and cell survival [36]. In addition, circRNAs may play key roles in the progression of various diseases through their biological effects, in part, by interacting with RBPs, serving as sponges of miRNA, and/or contributing to protein coding [37,38,39]. In fact, several circRNAs may be involved in regulating pathological processes [40]. Furthermore, circRNAs are abundant in the brain and/or in exosome vesicles [41]. Their capability to transverse the blood–brain barrier (BBB) makes them useful candidates as potential diagnostic tools for central nervous system (CNS) disorders [41]. Some circRNAs’ expression increases during CNS development to raise the concentration of miRNA target sites [30]. Remarkably, a large proportion of circRNAs are abundant in the brain with an unequal distribution in the neuronal compartments. CircRNAs may be involved in the regulation of circulating miRNA genes in CNS disorders such as Alzheimer’s disease [42]. Furthermore, circRNAs have been recognized as potential biomarkers in other diseases including amyotrophic lateral sclerosis (ALS), diabetes, and glioblastoma [43]. For example, *circSMOX* RNA has been identified as a biomarker in genetic mice models of ALS with the potential for indicating disease progression [44].

Interestingly, circRNAs may decrease during cell proliferation, even in some cancer cells [45]. Many circRNAs with miRNA response elements have been discovered to play essential roles by acting as endogenous competitive RNAs [10]. For example, ciRS7, a typical sponge of miR-7, contains more than 70 miR-7 binding sites [37]. In addition, circSPARC may upregulate the expression of janus kinase 2 (JAK2) by competitively binding to miR-485-3p, and might augment the migration and/or invasion of colorectal cancer [46]. In addition, circRNAs have been revealed to accomplish biological functions by interacting with RBPs and/or participating in protein coding [47]. Furthermore, circRNAs may also regulate transcription to disturb the expression of their parental genes [48]. Amazingly, the unique structure of circRNAs makes them exciting for use as potential diagnostic biomarkers even for cardiovascular diseases [49]. Likewise, recent evidence has identified a crucial role of several extracellular circRNAs in alleviating damage due to cardiohypertrophy, heart failure, and myocardial infarction [50]. Furthermore, several studies have reported their association with inflammatory responses; thus, influencing pathophysiological phenomena in various tissues [51], which may become targets for disease therapy. CircRNA is an important cargo carried by exosomes, which could modulate gene expression by sponging certain miRNAs, regulating nuclear transcription, and competing with mRNA splicing [52]. In addition, their closed loop structure determines the high biological stability of circRNAs; thus, making them a promising biomarker in clinics.

## 3. CircRNAs and Several Diseases

CircRNAs have attracted general interest for their stability, abundance in the brain, capability to cross the BBB, and their specific expression in several diseases. Accumulating evidence has shown that aberrant expression of circRNAs could play a key role in the pathogenesis of several diseases. Notably, circRNAs might be indispensable immune system gene modulators, which might be strongly linked to the occurrence of autoimmune disorders. Here, we present the current findings and theragnostic potentials of circRNAs in common diseases. This section aims to provide new insights to support the development of novel diagnostic and/or therapeutic strategies for these diseases (Figure 1).

### 3.1. CircRNAs and Cancers

CircRNAs have been found in exosome vesicles, where they are thought to modulate the expression of several genes and miRNAs, which has gained increased attention in cancer research [53,54,55]. Some circRNAs could promote a malignant phenotype of peripheral tumor cells in cholangiocarcinoma [56]. Several studies on circRNAs have revealed the involvement of circRNAs in glioma progression by competitive sponging of miRNAs [57]. For instance, *circ_0037655* may be able to enhance gliomas to progress via the direction of miR-214 and phosphoinositide-3 kinase (PI3K)/AKT signal transduction [58]. As mentioned before, circRNAs are structurally stable, presumably because their lack of both 5′ and 3′ ends might be resistant to exonuclease activity, which might enable circRNAs to serve as diagnostic and/or prognostic biomarkers for cancers. Furthermore, some circRNAs may play important roles in offering potential therapeutic targets [59]. A correlation between circRNAs and pancreatic cancer has also been reported [60]. For example, the silencing of circ_0030235 by an siRNA principally suppresses the cell proliferation, migration, and/or invasion of pancreatic cancer [60]. Similarly, circRNA 100146 may employ its oncogenic effect on non-small cell lung carcinoma by interacting with miR-361-3p [61]. In addition, circSMARCA5 may be aberrantly expressed in a variety of diseases, which could be shown to have prognostic value in malignant tumor cells [62]. Interestingly, circRNA_000864, miR-361-3p, and BTG2 could function as potential targets for the treatment of pancreatic cancer [63]. The overexpression of c*ircBTG2* might inhibit the proliferation and/or invasion of glioma cells, whereas *circBTG2* knockdown could promote tumor growth in vivo [64]. *CircBTG2* may repress miR-25-3p to prevent it from interacting with some RNAs in other pathways [64]. Another member of the APRO circRNAs, circRNA BTG3, has been revealed to facilitate the proliferation of colorectal cancer cells and/or lung cancer cells [65,66]. BTG3, a known antiproliferative protein, has been shown to be a direct target of miR-106b-5p, whose expression level may be reverse correlated with miR-106b-5p expression [67]. Gastric cancer-derived exosomal miR-552-5p could also facilitate tumorigenesis by interfering with the phosphatase and tensin homolog (PTEN) and/or the Tob1 signaling axis [68]. Both PTEN and Tob1 have been shown to be expressed at extraordinary levels in adjacent non-cancerous tissues, while miR-552-5p may be expressed at lower levels there [68].

### 3.2. CircRNAs and CNS Disorders

CircRNAs are commonly expressed in the nervous system, particularly in the brain [69]. The specific expression of some circRNAs in the brain identifies these as candidates for biomarkers of neurodegenerative diseases. For example, there are several circRNAs that are differentially expressed in the brain tissue and plasma of patients with Alzheimer’s disease [70]. In addition, studies have identified a few circRNAs in neurodegenerative diseases such as multiple system atrophy [71]. There are also a few reports on circRNAs in Parkinson’s disease [72]. Furthermore, several reports have suggested a link between circRNAs and ALS [73]. These are involved in important signal transduction pathways in the regulation of various neural activities [74]. The fact that circRNAs are abundant in the brain and exosomes could make them beneficial biomarkers for CNS disorders [41,75]. The circRNAs are enriched by more than two-fold in exosomes compared to those retained in the cells, which may provide some information about the disease status of CNS disorders and/or brain tumors including glioblastoma [41,75], suggesting that circRNAs may be involved in regulating the biological activity of brain cells [75]. Upregulating circPAIP2, an intron-retained circRNA, may affect translational inhibition of memory-related genes through the reactivation of PABP, which might be associated with the role of RNA polymerase II localized in the nuclei [13,76,77]. It has been reported that RNA binding proteins such as PABP may co-localize with small tau protein inclusions in Alzheimer’s disease. APRO family proteins may be also involved in the translational suppression of mRNAs through their interaction with PABP [78]. In addition, it has been suggested that the miR-146a-mediated suppression of BTG2, another member of the APRO family, might contribute to the protective role of neurons in postoperative cognitive dysfunction [79]. Furthermore, it has been shown that *BTG3* may be implicated in neurogenesis [80].

### 3.3. CircRNAs and Diabetes Mellitus

Accumulating evidence has proved that circRNAs are associated with some diabetes. Some studies have identified the circRNA expression profiles in type 1 diabetes mellitus (T1DM), indicating that circRNAs might play a key role in the progression of T1DM [52,81]. For example, a differential expression profile of plasma circRNAs such as hsa_circRNA_100332, hsa_circRNA_101062, hsa_circRNA_103845 and/or hsa_circRNA_085129 may be possibly associated with the onset of T1DM [52]. In particular, exosomes with circRNAs might participate in the progression of T1DM via multiple mechanisms. Therefore, the biomarker potential of exosomes in T1DM has been emphasized. For example, it has been indicated that T lymphocyte exosomes could trigger beta-cell apoptosis via exosomal miRNAs [82]. In addition, the pancreatic islets could release the intracellular autoantigens of the pancreatic beta-cell into exosomes, which could be engaged by antigen-presenting cells for autoimmune disorders [83]. In general, exosomal RNAs derived from human islets may be expressed under the treatment of proinflammatory cytokines, emphasizing the biomarker potential of exosomal RNAs [84]. In fact, several exosomal mRNAs may be related to the progression of T1DM [85,86]. Furthermore, miR-21-5p in circulating exosomes has been increased during the development of T1DM [87]. In addition, it has been suggested that exosomes released by adipose tissue-derived mesenchymal stem cells (MSCs) may possess immunomodulatory effects on T lymphocytes, which could improve the symptoms of T1DM [88]. In addition, the role of circRNAs in the occurrence and/or development of type 2 diabetes mellitus (T2DM) have also been shown. For example, hsa_circ_CCNB1 and/or hsa_circ_0009024 could be utilized as possible biomarkers for T2DM [89]. CircTulp4 may stimulate cell cycle progression, thereby discharging INS-1 cell dysfunction under glycoside toxicity [90]. In these ways, circRNAs may also participate in the occurrence and/or development of T2DM through various mechanisms with circRNAs. Furthermore, several circRNAs have been reported to be dysregulated in gestational diabetes mellitus [91]. Moreover, some of them may be of noteworthy diagnostic value even for T2DM [92]. Here too, it has been reported that circRNAs may regulate gene expression by controlling the functions of miRNAs, RBP, and/or PABP [93], which may be followed by binding to the 5′-untranslated region (UTR) of insulin mRNA to increase the protein translation and/or the secretion of insulin in β cells [94].

### 3.4. CircRNAs and Bone Related Diseases

Attention has been paid to the role of circRNAs in osteogenic differentiation [95]. CircRNAs have been reported to influence the differentiation of osteoblasts and/or osteoclasts [96,97], which might be deeply involved in the regulation of bone metabolism [96,98]. For example, circRNA_009934 is particularly expressed during osteoclast differentiation, which could attach miR-5107 to promote the expression of tumor necrosis factor receptor-associated factor 6 (TRAF6) [99]. On the contrary, the expression of circ_0021739 could inhibit osteoclast differentiation via the targeting of miR-502-5p [100]. Circ_0024097 could also promote osteogenic differentiation by binding to miR-376b-3p [101]. In addition, circ_0076906 could promote the expression of osteoglycin before inducing the differentiation of bone marrow derived mesenchymal stem cells (BMSCs) into osteoblasts [102]. Circ_0006215 could also bind to miR-942-5p to encourage the differentiation of BMSCs into osteoblasts by regulating runt-related transcription factor 2 (RUNX2) and/or vascular endothelial growth factor (VEGF) expression [103]. Therefore, the overexpression of circRUNX2 could help the osteogenic differentiation to move away from the progression of osteoporosis [104]. Consistently, circRNA-fibroblast growth factor receptor 2 (Fgfr2) could sponge miR-133 to regulate the expression of bone morphogenetic protein-6 (BMP6) for osteogenesis [105]. Similarly, circRNA-23525 could also promote osteogenic differentiation by sponging miR-30a-3p to regulate RUNX2 expression [106]. In the meantime, however, circ_0011269 could regulate the expression of RUNX2 to enhance the progression of osteoporosis [107]. APRO family proteins may also be involved in osteogenesis. For example, miR-26a exerts its effect by directly targeting Tob1, the negative regulator of the BMP/Smad signaling pathway, by binding to the 3′-untranslated region with PABP and, thus, repressing Tob1 protein expression [108]. In addition, Tob2 could also inhibit the formation of osteoclasts by interacting with the vitamin D receptor (VDR) to suppress the expression of the receptor activator of the nuclear factor-kappa B ligand (RANKL) [109]. BTG2 has been shown to be significantly downregulated in the cartilage of osteoarthritis animal models [110].

### 3.5. Polycystic Ovary Syndrome

CircRNAs are also associated with follicular development, ovarian senescence, spermatogenesis, and/or the process of germ cell development, suggesting that circRNAs might function in the regulation of several germ cells [111]. Polycystic ovary syndrome (PCOS) is a complex metabolic disorder seen in females of reproductive age. The pathology of PCOS may be multifactorial dysfunction in several pathways including ovarian folliculogenesis, gonadotropin production, and/or gut microbiota imbalance [112]. The cause of PCOS may also be affected by environmental factors [113]. Low-grade inflammatory conditions, such as obesity, with elevations of inflammatory cytokines, may be common metabolic disorders in women with PCOS. Interestingly, it has been reported that PCOS is related to cardiovascular risk factors [114]. The incidence of PCOS in premenopausal women has been reported to be 6–20%, and it might be the most common endocrine disease in adult women [115]. There is no test to conclusively diagnose PCOS [116]. The role and/or mechanism of circRNAs in PCOS have gradually become a research hotspot [117]. Remarkably, the circRNAs have been shown to be mainly enriched in the AGO2 complex in PCOS [118]. The other circRNAs including circPUM1, could promote the progression of PCOS through sponging of miR-760 [119].

## 4. Mechanism of circRNAs’ Action with PABP and APROs

As shown here, a lot of evidence suggests that circRNAs may play a key role in disease initiation and/or progression. In addition, circRNAs may play a key role in the proliferation, differentiation, and/or apoptosis of various cells in those diseases. Given the diverse roles attributed to circRNAs and miRNAs, the molecular biological mechanisms of these RNAs might be quite elaborate. Again, circRNA could regulate gene transcription, alternative splicing, molecular sponges of miRNA, RNA-binding proteins, and/or protein translation. In translation, miRNAs might be bound with AGO proteins in the 3′ UTR of complementary mRNA sites. The AGO-miRNA activity might be further modulated by adjacent RBPs by interacting with target proteins. A single circRNA can bind to one or more miRNAs through its circular sequences. Accordingly, circRNAs could also interact with transcription machinery including RNA Pol II and/or U1 snRNP to promote their parent gene expression in the nuclei. Probably, long non-coding RNA could act as a competitive endogenous RNA to compete for miRNA binding. Some p-element induced wimpy testis (piwi)-interacting RNAs could also interact with BTG1 expression [120]. Piwi proteins have been shown to be a subfamily of Argonaute proteins that maintain germ cells in eukaryotes. The knockdown of piwil1, a member of the piwi-like protein family, could increase the expression of the transcriptional co-regulator BTG2 [121]. Piwi-interacting RNAs and PIWI proteins may be essential in cells to repress transposons and/or to regulate mRNAs. Consequently, circRNAs could regulate mRNA stability and immune cell death by binding to specific RBPs. In addition, circRNAs could indirectly regulate gene expression by binding targeted miRNAs in cells including immune cells [122]. Certain circRNAs might also block the translation of the host gene by binding to the adjacent PABP [123]. The association of circRNAs and PABP might affect the combination of PABP and eIF4G on the 5′ cap region of mRNA, which specifically affects the translation and/or the expression of certain mRNA [123] (Figure 2).

Some circRNAs are subject to endoribonucleolytic cleavage with the target of mRNA and/or miRNA [124]. Several members of the APRO family are also shown to be implicated in cytoplasmic mRNA deadenylation and its turnover [125]. The N-terminal conserved APRO domain is capable of binding to DNA-binding transcription factors as well as the deadenylase subunits of the CCR4/NOT complex [126]. Likewise, some of the APRO family proteins could interact with the poly(A) nuclease complex CCR4/CAF1 and the cytoplasmic PABP [25,127]. In addition, it has been shown that the Tob1 and Tob2 proteins contain an extra-long C-terminal domain with two PAM2 motifs [128]. These APRO proteins (Tob1 and Tob2) could interact with CAF1 and PABP simultaneously, which might stimulate the deadenylation of mRNA [127]. Interestingly, the antiproliferative effects of Tob1 have been suggested to be involved in the exploitation of the CAF1/CCR4 deadenylase complex [129], suggesting that APRO proteins could exert their antiproliferative activity by modulating the turnover of mRNA [130]. In fact, BTG2 could also interact with CAF1 deadenylase through its APRO domain to control cell proliferation [131]. It has been shown that mRNA destabilization by BTG1 and/or BTG2 may sustain cell quiescence [132]. Hence, circRNAs and miRNAs could inhibit mRNA expression by base-pairing to the 3′ UTR of the target mRNAs, which consequently inhibits translation by initiating poly(A) tail deadenylation and mRNA destabilization with APRO family proteins [133]. In fact, the miRISC could interact with PABP, CAF1, and CCR4 deadenylases [134]. Importantly, a core component of the miRISC could interact with the PABP and APRO family proteins, which may be compulsory for the miRNA-mediated deadenylation [134]. Since the APRO family proteins have the potential to interact with the CCR4/CAF1 complex, APRO family proteins could be a key modulator of circRNAs- and/or miRNAs-function. Therefore, APRO and the CCR4/CAF1 protein complex might be a multifunctional regulator that plays an important role in multiple cellular processes in eukaryotes [135]. The expression of APRO family proteins may be also regulated by certain circRNAs and/or miRNAs [120,136].

## 5. Future Perspectives

Non-coding RNAs are involved in the regulation of diverse cellular processes, including transcription, RNA processing, translation and genome organization. In particular, circRNAs may play crucial roles in cancers, nervous system diseases, immune diseases and metabolic diseases. In detail, circRNAs could regulate gene expression, RNA binding protein interactions, and polymerase II transcription regulation by miRNA sponge activity. Certainly, circRNAs have attracted general interest for their stability and their highly tissue-specific expression. Given the growing interest in RNA biomarkers for several diseases, circRNAs could represent reliable and affordable candidates. First, their circular structure endows them with high RNase resistance and with peculiar structural conformations unlike linear RNAs. They are present in the blood circulation supporting their possible usefulness as disease biomarkers. For example, the observation that differentially expressed circRNAs in the brain overlaps with those in the plasma of patients affected with neurodegenerative diseases has led to the promising perspective of their potential use as peripheral biomarkers [137]. Differentially expressed blood circRNAs may also be novel inflammatory biomarkers. The mechanism of circRNAs with RBPs, including the APRO family proteins, for disease progression needs further investigation. Their differential expression in disease-associated genes suggests that they also represent crucial determinants of pathophysiological processes implicated in those diseases. A deeper understanding of their molecular mechanisms in physiological as well as pathological conditions should remain warranted. It is necessary to study the function of circRNAs and circRNA-binding proteins further, which will help us to integrate circRNAs into the treatment of related diseases as well as provide new promising therapeutic and diagnostic approaches.

## 6. Conclusions

CircRNAs and several RNA-binding proteins including APRO family molecules could be crucial modulators for gene expression, which might be linked to the pathogenesis of various diseases.

## Figures and Tables

**Figure 1 ncrna-09-00023-f001:**
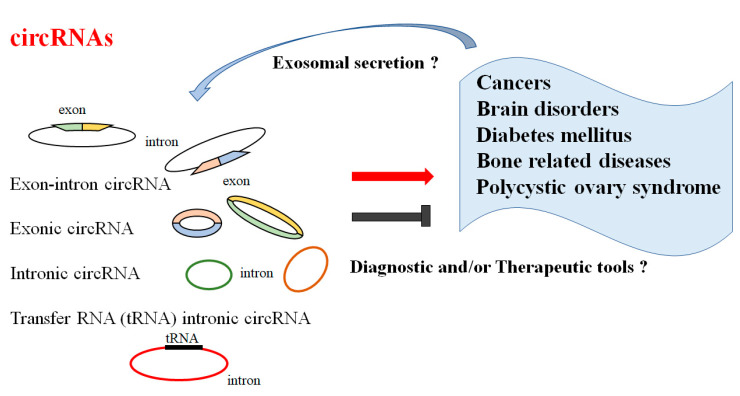
Illustration of the general functions of circular RNAs (circRNAs). Functions of circRNAs have been proposed in several diseases including cancers, brain or CNS disorders, several types of diabetes mellitus, bone-related diseases, and polycystic ovary syndrome. Consequently, certain circRNAs could be diagnostic and/or therapeutic tools for these diseases. The arrowhead means stimulation and/or augmentation, whereas the hammerhead represents inhibition.

**Figure 2 ncrna-09-00023-f002:**
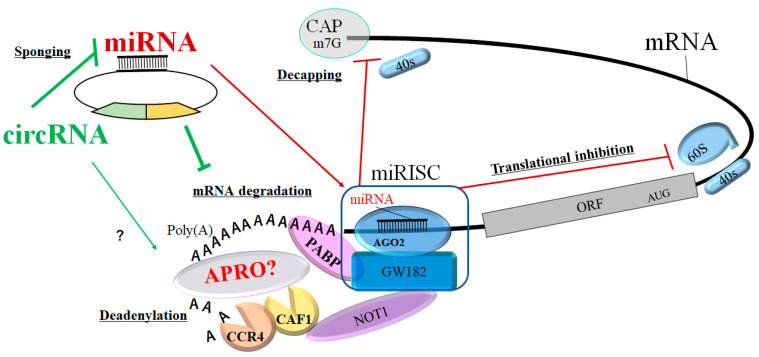
Schematic representation of circRNA- and/or miRNA-mediated functional inhibition of mRNA translation. The AGO2 protein interacts with GW182 constructing the miRNA-loaded RNA-induced silencing complex (miRISC), which may facilitate the deadenylation and/or mRNA degradation process by CAF1/CCR4/NOT1 with the PABP and APRO protein complex. Consequently, the circRNA and/or miRNA could play an active role in regulating post-transcriptional gene expression via the decapping, translational inhibition, deadenylation, and degradation of mRNA. The CAF1/CCR4/NOT1 complex is recruited to the 3′ UTR of specific mRNAs through an interaction with the PABP protein. APRO family proteins might also interact with PABP to recruit the CAF1/CCR4/NOT1 complex. The arrowhead means stimulation, and the hammerhead represents inhibition. Note that some critical pathways have been omitted for clarity. Abbreviations: ORF, open reading frame; miRISC, microRNA-induced silencing complex, “?” means for our speculation.

## Data Availability

Not applicable.

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
