# Peer review of "CircRNAs and RNA-Binding Proteins Involved in the Pathogenesis of Cancers or Central Nervous System Disorders"

_ncrna, 2023, doi:10.3390/ncrna9020023_

Round 1
Reviewer 1 Report
The authors reviewed a broad range of literatures in the field of circular RNAs related to diseases, diagnosis, and their interactions with a variety of proteins and RNAs. The highlight of this review paper is that they summarized current findings of circular RNAs in each type of disease. In each subsection, the authors were able to provide adequate references of recent studies within the field of molecular biology research of circular RNAs. However, there are some structural issues and specific question for some paragraphs that will need the authors to clarify as listed below.
1. The authors had collected many studies for each disease. It is recommended that the author can make a summarized table for each disease to lay out their primary circular RNAs, their interaction targets (miRNAs, proteins, mRNAs, etc.), any known mechanisms, any known downstream effects, and their reference numbers. This will greatly help the readers to quickly get a big picture of this review paper and find the topics that they are interested in more easily.
2. The Section 2 (Roles of circRNAs in pathophysiology) is relatively unclear. First, the structure of this section is mixed with a variety of different diseases. There is no clear connection between those diseases, but they are all written together. Secondly, if this section is emphasizing the function of circular RNAs as a biomarker, can the authors explain why these specific circular RNAs can be the biomarker for diagnosis of each disease? (Not circular RNAs as a biomarker in general, but specific to each disease – what does the mentioned circular RNAs do?)
3. For reference [1] in lines 27-28, can the author point out where did Dr. Hwang indicated “abnormal expressions of non-coding RNAs”? Feel free to directly quote the sentences from Dr. Hwang’s paper to respond this question.
4. Figure 1 is redundant and add no additional information to the paper. It can be replaced by a summarized table like aforementioned. Feel free to justify your rationale of this figure if the authors disagree.
5. In lines 208-223, the author summarized the potential mechanism of circRNAs to T1DM. Can the authors provide any specific circular RNAs that has been studied for T1DM?
6. BTG2/3 seems to be part of many diseases including pancreatic cancer, colorectal cancer, neuro/cognitive dysfunction, osteoarthritis, and more. Can the authors explain this?
7. In the Future perspectives section, in addition to all the great potentials of circular RNAs, the authors are encouraged to point out any limitations or challenges of the current research based on their overall review in this field.
8. Minor formatting or typos:
a. Please check all the words are in a same font and size. For example, line 158 “0037655” “via”, line 173 “_CircBTG2”, line 203 “In addition”. And some reference numbers are bigger than the others.*
b. Please carefully proofread the manuscript and correct all the grammatical and typographical errors. Some conjunction words are unnecessary.*
c. Line 174, “Another member of the member of”, please make sure this is not a duplication.
d. Please check all the acronyms to make sure that they were only spell out in their full term as they first appear in the manuscript. For example, line 258, BTG2.
e. Line 291, “contend”? line 345 “downbeat”?
*Note that this is not an exhaustive list. Please go over the article and correct all the errors.
Author Response
Reviewer1
The authors reviewed a broad range of literatures in the field of circular RNAs related to diseases, diagnosis, and their interactions with a variety of proteins and RNAs. The highlight of this review paper is that they summarized current findings of circular RNAs in each type of disease. In each subsection, the authors were able to provide adequate references of recent studies within the field of molecular biology research of circular RNAs. However, there are some structural issues and specific question for some paragraphs that will need the authors to clarify as listed below.
- The authors had collected many studies for each disease. It is recommended that the author can make a summarized table for each disease to lay out their primary circular RNAs, their interaction targets (miRNAs, proteins, mRNAs, etc.), any known mechanisms, any known downstream effects, and their reference numbers. This will greatly help the readers to quickly get a big picture of this review paper and find the topics that they are interested in more easily.
Exactly, you are right. That is a point really! However, it is very difficult for us to summarize or construct them at this stage. Please allow us to provide them in the next coming paper.
- The Section 2 (Roles of circRNAs in pathophysiology) is relatively unclear. First, the structure of this section is mixed with a variety of different diseases. There is no clear connection between those diseases, but they are all written together. Secondly, if this section is emphasizing the function of circular RNAs as a biomarker, can the authors explain why these specific circular RNAs can be the biomarker for diagnosis of each disease? (Not circular RNAs as a biomarker in general, but specific to each disease – what does the mentioned circular RNAs do?)
Of course, specific circular RNA might be a biomarker to the related disease. As shown here, APRO family proteins could employ their voluminous functions via the numerous kinds of circRNAs and/or miRNAs against the pathogenesis of various kinds of diseases including CNS disorders as well as systemic malignancies. According to this suggestion, however, we have improved the content of section 2.
- For reference [1] in lines 27-28, can the author point out where did Dr. Hwang indicated “abnormal expressions of non-coding RNAs”? Feel free to directly quote the sentences from Dr. Hwang’s paper to respond this question.
As we do not know where the point and/or the real meaning of this question are, reference [1] has been replaced with the latest one.
- Figure 1 is redundant and add no additional information to the paper. It can be replaced by a summarized table like aforementioned. Feel free to justify your rationale of this figure if the authors disagree.
Our rationale is for clarity and/or simplicity. However, all figures have been amended according to this suggestion,
- In lines 208-223, the author summarized the potential mechanism of circRNAs to T1DM. Can the authors provide any specific circular RNAs that has been studied for T1DM?
A sentence denoting “A differential expression profile of plasma circRNAs such as hsa_circRNA_100332, hsa_circRNA_101062, hsa_circRNA_103845 and hsa_circRNA_085129 may be potentially associated with the onset of T1DM” has been inserted within those lines.
- BTG2/3 seems to be part of many diseases including pancreatic cancer, colorectal cancer, neuro/cognitive dysfunction, osteoarthritis, and more. Can the authors explain this?
APRO family members including the BTG2/3 have been known as tumor suppressors. Another well-known tumor suppressor, p53, may be also involved in the pathogenesis of a variety of cancers as well as distinct disorders. As for BTG2/3, they could employ their voluminous functions via the numerous kinds of circRNAs and/or miRNAs based on the concept shown here.
- In the Future perspectives section, in addition to all the great potentials of circular RNAs, the authors are encouraged to point out any limitations or challenges of the current research based on their overall review in this field.
That is a point, again. It is too difficult for us to summarize it at this stage. Please allow us to provide it in the coming paper.
- Minor formatting or typos:
- Please check all the words are in a same font and size. For example, line 158 “0037655” “via”, line 173 “_CircBTG2”, line 203 “In addition”. And some reference numbers are bigger than the others.*
- Please carefully proofread the manuscript and correct all the grammatical and typographical errors. Some conjunction words are unnecessary.*
- Line 174, “Another member of the member of”, please make sure this is not a duplication.
- Please check all the acronyms to make sure that they were only spell out in their full term as they first appear in the manuscript. For example, line 258, BTG2.
- Line 291, “contend”? line 345 “downbeat”?
*Note that this is not an exhaustive list. Please go over the article and correct all the errors.
These typos have been amended. Thank you so much. In addition, again we have gone over the text/abstract and amended typos and grammatical errors as much as possible to improve the manuscript more helpful to the readers.
Reviewer 2 Report
1. Abstract is to the point, very good written.
2. Point wise explained each and everything is very good.
3. Just lacking the conclusion lines separately, else remain things are okay.
4. Figures are good just needed little alignment.
5. Diabetes explanation was too good in context of circular RNA
Author Response
Reviewer2
- Abstract is to the point, very good written.
Thank you so much for the good evaluation to the manuscript.
- Point wise explained each and everything is very good.
Again, thank you so much for the good evaluation.
- Just lacking the conclusion lines separately, else remain things are okay.
Accordingly, “6. Conclusion” section has been added to summarize the main idea of this manuscript at the last position.
- Figures are good just needed little alignment.
All figures have been amended.
- Diabetes explanation was too good in context of circular RNA
Thank you so much for the good evaluation to the manuscript. Explanation for the specific circular RNAs studied for T1DM has been appended.
Reviewer 3 Report
This manuscript first introduced the formation, structural properties and functions of circular RNAs (circRNA). Then the authors discussed circRNA-related diseases, such as cancers, brain disorders, etc, and reviewed the mechanism of cirRNAs action with RNA-binding-proteins (RBP). At last, the manuscript summarized the future perspectives of circRNAs regarding their usage for diagnostics and therapeutics.
While the text of the manuscript is fine, the figures need substantial improvement. In Figure 1, circRNAs are simply represented by ovals, and related diseases are listed with text. The stimulation and inhibition are represented in arrowhead and hammerhead, prospectively. The whole figure is not professional and informative. The same issue exists in Figure 2. Please refer to related review paper and refine the figures to be qualified for research papers.
Two statements in Introduction is not accurate. (1) it is said “circRNAs are much more stable than linear mRNAs or non-coding miRNAs [7]”. Citation [7] did not compare the stability between circRNA and miRNA. (2) it is said “In fact, it has been shown that circRNAs are more resistant than linear mRNA to the degradation of RNase R due to the lack of terminal 5′ caps and 3′ poly(A) tails [9].” CircRNA is more stable due to the fact that it does not have 5’ and 3’ ends. Linear RNA can also have no 5’ caps and 3’ poly(A) tails, which leads to even more instable.
Minor typos: (1) line 16: micro-RNAs -> microRNAs; (2) line 194: twofold -> two-fold; (3) line 280: may plays -> may play; (4) line 285—286: Ago/ago -> AGO.
Author Response
Reviewer3
This manuscript first introduced the formation, structural properties and functions of circular RNAs (circRNA). Then the authors discussed circRNA-related diseases, such as cancers, brain disorders, etc, and reviewed the mechanism of cirRNAs action with RNA-binding-proteins (RBP). At last, the manuscript summarized the future perspectives of circRNAs regarding their usage for diagnostics and therapeutics.
While the text of the manuscript is fine, the figures need substantial improvement. In Figure 1, circRNAs are simply represented by ovals, and related diseases are listed with text. The stimulation and inhibition are represented in arrowhead and hammerhead, prospectively. The whole figure is not professional and informative. The same issue exists in Figure 2. Please refer to related review paper and refine the figures to be qualified for research papers.
According to this suggestion, all figures have been amended.
Two statements in Introduction is not accurate. (1) it is said “circRNAs are much more stable than linear mRNAs or non-coding miRNAs [7]”. Citation [7] did not compare the stability between circRNA and miRNA. (2) it is said “In fact, it has been shown that circRNAs are more resistant than linear mRNA to the degradation of RNase R due to the lack of terminal 5′ caps and 3′ poly(A) tails [9].” CircRNA is more stable due to the fact that it does not have 5’ and 3’ ends. Linear RNA can also have no 5’ caps and 3’ poly(A) tails, which leads to even more instable.
Citation [7] has been replaced with the new one, in which the comparison of the stability between circRNA and linear RNA are described. According to the second indication, the sentence has been improved.
Minor typos: (1) line 16: micro-RNAs -> microRNAs; (2) line 194: twofold -> two-fold; (3) line 280: may plays -> may play; (4) line 285—286: Ago/ago -> AGO.
These typos have been amended. Thank you so much. In addition, again we have gone over the text/abstract and amended typos and grammatical errors as much as possible to improve the manuscript more helpful to the readers.
Round 2
Reviewer 1 Report
The authors have satisfactorily addressed all the comments.
Reviewer 3 Report
The quality of the manuscript is improved.